# Data Harmonization for Heterogeneous Datasets: A Systematic Literature Review

**Ganesh Kumar** [1,*], **Shuib Basri** [1], **Abdullahi Abubakar Imam** [1,2], **Sunder Ali Khowaja** [3], **Luiz Fernando Capretz** [4] **and Abdullateef Oluwagbemiga Balogun** [1,5]

1  Computer and Information Science Department, Universiti Teknologi PETRONAS, Bandar Seri Iskandar 32610, Perak, Malaysia; shuib_basri@utp.edu.my (S.B.); abdullahi_g03618@utp.edu.my (A.A.I.); abdullateef_16005851@utp.edu.my (A.O.B.)
2  Department of Computer Science, Ahmadu Bello University, Zaria 1044, Nigeria
3  Department of Telecommunication Engineering, Faculty of Engineering and Technology, University of Sindh, Jamshoro 76090, Pakistan; sandar.ali@usindh.edu.pk
4  Department of Electrical and Computer Engineering, Western University, London, ON N6A 5B9, Canada; lcapretz@uwo.ca
5  Department of Computer Science, University of Ilorin, Ilorin 1515, Nigeria
*  Correspondence: ganesh_17005106@utp.edu.my

**Abstract:** As data size increases drastically, its variety also increases. Investigating such heterogeneous data is one of the most challenging tasks in information management and data analytics. The heterogeneity and decentralization of data sources affect data visualization and prediction, thereby influencing analytical results accordingly. Data harmonization (DH) corresponds to a field that unifies the representation of such a disparate nature of data. Over the years, multiple solutions have been developed to minimize the heterogeneity aspects and disparity in formats of big-data types. In this study, a systematic review of the literature was conducted to assess the state-of-the-art DH techniques. This study aimed to understand the issues faced due to heterogeneity, the need for DH and the techniques that deal with substantial heterogeneous textual datasets. The process produced 1355 articles, but among them, only 70 articles were found to be relevant through inclusion and exclusion criteria methods. The result shows that the heterogeneity of structured, semi-structured, and unstructured (SSU) data can be managed by using DH and its core techniques, such as text preprocessing, Natural Language Preprocessing (NLP), machine learning (ML), and deep learning (DL). These techniques are applied to many real-world applications centered on the information-retrieval domain. Several assessment criteria were implemented to measure the efficiency of these techniques, such as precision, recall, F-1, accuracy, and time. A detailed explanation of each research question, common techniques, and performance measures is also discussed. Lastly, we present readers with a detailed discussion of the existing work, contributions, and managerial and academic implications, along with the conclusion, limitations, and future research directions.

**Keywords:** data harmonization; heterogeneous data; text preprocessing

## 1. Introduction

Big Data play a vital role in the assessment of massive data produced every second by real-world applications, using tools and algorithms [1]. Some of real-life application domains of Big Data are healthcare, telecommunication, financial firms, retail, law enforcement, marketing, new product development, banking, energy and utilities, insurance, education, agriculture, and urban planning, as discussed in Reference [2]. Nowadays, data are being produced in various formats, ranging from structured and semi-structured to unstructured (SSU) generated from heterogeneous resources [3,4]. The disparate nature of data cannot be processed with simple tools and techniques [2,5], and this

creates a challenge for decision-makers to make decisions based on the scattered data. Emerging technologies, such as the Internet of Things (IoT), Industry 4.0 (I4.0), and extended reality (XR), produce distinct kinds of information via heterogeneous sources and real-world applications that create heterogeneity issues [6], in IoT integration, security, analytics challenges, and computational time [7–9]. Among them, data harmonization (DH), which describes the uniform representation of heterogeneous data, was proposed in References [10,11].

IoT is a system that deals with interrelated computing objects, such as unique tags, RFID, or machine interactions, and that can transfer data without human and machine involvement [12]. As technology evolves, the IoT has further grown into the Industrial IoT (IIoT), which deals with heterogeneous data produced by real-world applications, industrial products, and devices, such as privacy authentication logs of IIoT devices [13], business architecture devices data [14], and heterogeneous IIoT devices data [6]. In addition, I4.0 deals with IoT-based automation, technologies, and decision-making that help decision-makers to make decisions based on the disparate nature of data produced [15]. Applications of I4.0 in higher education, predictive maintenance [16,17], food logistics [18], knowledge management [19], business [20], and supply chain [21]. The main problem faced by these applications is related to managing the heterogeneous data produced in bulk by employing I4.0 and IIoT. The data produced by industries include digital data for manufacturing purposes, unstructured data for predictive maintenance, customer data for food logistics, customer reviews for knowledge management, business data for the supply chain, and manufacturing data for the supply chain. XR deals with the real and virtual environment with the help of a machine and human interaction [22]. XR is improving heterogeneous manufacturing data in the digital world. The tools must be advanced so that user acceptance and better usability of products are achieved [23]. AI can be effectively used to address the disparate nature of manufacturing data to deliver the best appearance to the XR industry [24].

To resolve the problems mentioned earlier, the disparity of data needs to be reviewed in detail, so that data harmonization models, tools, techniques, algorithms, and their performance can be evaluated for extensive heterogeneous textual information. Although related work was carried out in multimodalities for text, image, audio, and video [25–27], there were no such studies highlighting the work associated with textual data, data harmonization core techniques, and performance measurement. Multiple studies have been conducted which deal with applications such as sentiment analysis, text similarity, word embedding, and emotion recognition in conjunction with the help of classification and clustering techniques. Therefore, solving real-world application problems, such as those of a medical and healthcare nature, needs data to be harmonized and uniformly presented, so that decisions can be carried out efficiently. Based on the needs and contributions of emerging technologies and real-world application domains, we aimed to conduct a systematic review of the literature that could demonstrate the heterogeneity issues faced by real-world applications, data harmonization as a solution architecture for the disparate nature of data, techniques that can deal with large textual heterogeneous datasets, and performance assessment of models.

In this SLR, we have proposed to solve the problems mentioned above by doing a systematic review of the literature on textual data domains that deal with heterogeneity. The disciplines added were data integration, curation, and harmonization and were performed and contributed by the research community in their novel ideas. In addition, we selected articles that deal with the core textual data techniques and performance measurement techniques. The state-of-the-art SLR contains heterogeneity issues, textual data harmonization, data-processing techniques, and models' performance measurement methods. The research questions were drawn to emphasize the domains that focus on the heterogeneity issue, how the data harmonization approach will help, the core techniques that can deal with sizeable textual data, and which algorithms are suitable based on efficiency. The objectives of the proposed SLR are to understand the issues of heterogeneity faced by

industries; to mold the heterogeneity issue by replacing DH; and to keep an acceptable level of ML, DL, and NLP core techniques and performance measurement techniques for concerning sizeable textual data. This study will contribute towards Big Data variety and the data analytics research community that helps in representation, visualization, and prediction of the heterogeneous information produced in the disparate form.

This paper is organized as follows: In Section 2, an illustration of research methodology is presented that comprises three steps: planning, conducting, and reporting of the review. The results of selected articles, research questions, standard techniques, data formats, and performance techniques are discussed in Section 3. In Section 4, a discussion of existing solutions is presented, along with their contributions, managerial and academic implications, and the conclusion by including limitations, future research directions, and the scientific contribution of this review.

## 2. Research Methodology

In this Systematic Literature Review (SLR), the guidelines were followed from References [28,29]. The research process is divided into three phases. In the first planning phase, the stages of defining research questions, developing, and validating review protocols are covered. In the second phase, identification and selection of relevant studies, data extraction, and the information synthesis process are covered; and in the third phase, writing and validating the review are reported. Figure 1 illustrates the flow of three phases.

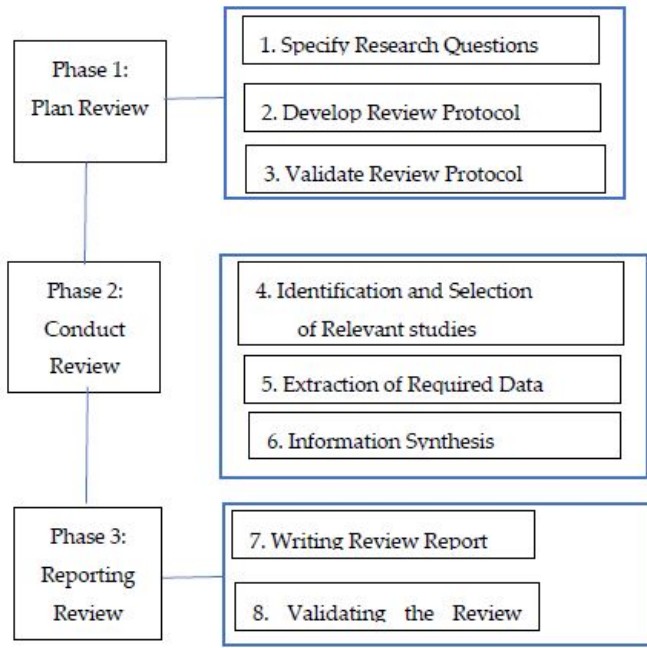

**Figure 1.** SLR process.

### 2.1. Plan Review

In this first phase of research methodology, the significant research questions and development of review protocols are specified with the proper searching strategy.

#### 2.1.1. Research Questions

In this SLR, the following research questions are set, and possibly all questions are later answered with proper solutions.

1.　RQ #1: Which of the domains are mainly focused on researching heterogeneity?

The motivation behind this research question is to find the significance of heterogeneity or heterogeneous data produced by industries, websites, and hospitals from models, frameworks, or applications. Heterogeneous data comprise structured, semi-structured, and unstructured, which is very tough to manage, and data are important for organizations, industries, and firms.

2.  RQ #2: How does data harmonization resolve the issues of heterogeneity?

This research question is linked to heterogeneity because we need to check the solution to it. The motivation for this research question is to check the models, frameworks, and applications that manage the disparate nature of data with the latest tools and techniques. Moreover, a different strategy was adopted to gather relevant data by using terms such as "data integration", "data mapping", and "data fusion".

3.  RQ #3: Which techniques are being used to solve the harmonization issue for large textual datasets?

The purpose of this research question is to identify the textual data and the techniques used for solving the issues of storing, managing, and uniform representation. Textual data can be used for semantic, syntactic, and schematic representation. Furthermore, it is used for predictive analysis and information retrieval. Thus, in-depth techniques for text data retrieval, data formats, and performance measures are later reviewed.

4.  RQ #4: Which deep learning algorithms are well-suited with respect to efficiency for large sequential datasets

In this research question, the main target is to identify the algorithms' performance for sequential text data processing. The textual data are completely based on heterogeneous data, data harmonization, and techniques.

### 2.1.2. Review Protocols

The development and validation of the review protocol highlight the searching of related articles with the appropriate keywords and the literature sources.

### 2.1.3. Searching Keywords

To guarantee that the review closely covers data harmonization and relevant techniques for heterogeneous data, we tried to limit our search to the most relevant search term. Thus, we started with the keywords, and then we went through the following steps:

- Extracting the major distinct terms from our research questions;
- Using different spellings of the terms;
- Updating our search terms with keywords from relevant papers.

We used the main alternatives and added "OR operator" and "AND operator" to get the maximum amount of directly relevant works in the literature, as shown in Table 1.

**Table 1.** Inclusion and exclusion criteria description.

| ID | Keywords |
|----|----------|
| 1 | "Data Harmonization" AND ("Heterogeneous Data" OR "Heterogeneity") AND ("Textual Data" OR "Text Data") AND ("text Preprocessing "OR "Preprocessing") |
| 2 | "Data Harmonization" AND ("Heterogeneous Data" OR "Heterogeneity") AND ("Textual Data" OR "Text Data") AND ("Text Preprocessing "OR "Preprocessing") AND (Techniques OR Algorithm) |
| 3 | "Data Integration" AND ("Heterogeneous Data" OR "Heterogeneity") AND ("Textual Data" OR "Text Data") AND ("text Preprocessing "OR "Preprocessing") |
| 4 | "Data Integration" AND ("Heterogeneous Data" OR "Heterogeneity") AND ("Textual Data" OR "Text Data") AND ("text Preprocessing "OR "Preprocessing") AND (Techniques OR Algorithm) |
| 5 | "Data Fusion" AND ("Heterogeneous Data" OR "Heterogeneity") AND ("Textual Data" OR "Text Data") AND ("text Preprocessing "OR "Preprocessing") |

| | |
|---|---|
| 6 | "Data Fusion" AND ("Heterogeneous Data" OR "Heterogeneity") AND ("Textual Data" OR "Text Data") AND ("text Preprocessing "OR "Preprocessing") AND (Techniques OR Algorithm) |
| 7 | ("Data Harmonization" OR "Data Integration" OR "Data Fusion") AND ("Heterogeneous Data" OR "Heterogeneity") AND ("Textual Data" OR "Text Data") AND ("text Preprocessing "OR "Preprocessing") |
| 8 | ("Data Harmonization" OR "Data Integration" OR "Data Fusion") AND ("Heterogeneous Data" OR "Heterogeneity") AND ("Textual Data" OR "Text Data") AND ("text Preprocessing "OR "Preprocessing") AND (Techniques OR Algorithm) |
| 9 | ("Data Harmonization" OR "Data Integration" OR "Data Fusion") AND ("Heterogeneous Data" OR "Heterogeneity") AND ("Textual Data" OR "Text Data") AND (Techniques OR Algorithm) |
| 10 | ("Data Harmonization" OR "Data Integration" OR "Data Fusion") AND ("Heterogeneous Data" OR "Heterogeneity") AND ("Textual Data" OR "Text Data") |

2.1.4. Literature Resources

- Primary review studies: Web of Science, Scopus, ACM Digital Library, Springer, Science Direct, and IEEE Explorer databases were chosen for selection of relevant articles. These databases have maximum coverage of quality articles in our domain, such as ISI and Scopus indexed articles. The search term was constructed by using the advanced search features provided by each of these databases. Our search included the period from 2015 to 2020.

*2.2. Conduct Review*

In this phase, we conducted the review according to the research questions, keywords, and protocols. This phase mostly emphasizes the inclusion and exclusion of articles, according to Table 2a, 2b.

**Table 2.** (**a**) Inclusion criteria description and (**b**) exclusion criteria description.

| **Inclusion Criteria** |
|---|
| The research was relevant to heterogeneous data sources. |
| The research was directly related to the data. |
| The research was related to text preprocessing and NLP applications. |
| The research used performance measurement techniques. |
| The research was conducted using ML, DL and NLP techniques related to textual data. |
| For duplicate publications of the same study, the newest and most complete one was selected. This is recorded for only one study whose related work appeared two times. |
| **Exclusion Criteria** |
| Studies that were irrelevant to data harmonization and domain were skipped. They showed up in our search due to the misuse of the term "harmonization" to describe traditional chemical and music work. Table A1 shows selected studies. |

2.2.1. Study Selection

The whole process of study selection is illustrated in Figure 2. A total of 1355 articles appeared in the online search. By applying filtration with title, keyword, inclusion, and exclusion criteria, a total of 155 papers were short-listed. Inclusion and exclusion criteria are defined in Table 2a, 2b. Among them, 33 articles were repeated in other databases, and 22 articles were from different domains, such as chemistry, music, and other languages. At the end, 30 articles are removed from the list after going through full reading.

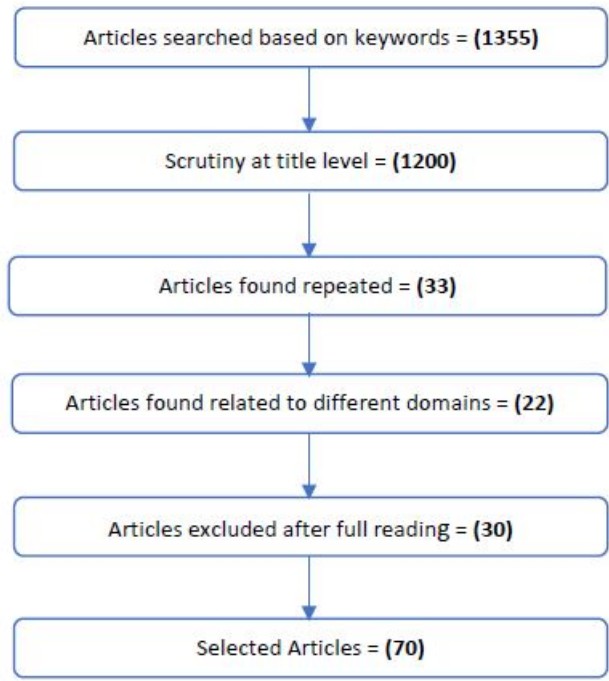

**Figure 2.** Process of identifying relevant studies.

Table 2a, 2b describes the selection criteria of the relevant articles according to keywords. The duplicate articles and articles that do not cover all research questions are excluded.

Table 3 illustrates the quality checklist questions for evaluation of studies. The questions are mainly designed for selection of the studies that are more relevant, detailed, and that cover all research questions.

**Table 3.** Quality checklist.

| No. | Questions |
|---|---|
| 1 | Did the studies focus on the heterogeneous nature of data? |
| 2 | Was the study explaining the harmonization of large textual data? |
| 3 | Was there any model proposed for textual data harmonization? |
| 4 | Is study focusing on the core techniques of ML, DL and NLP for large textual data? |
| 5 | Is the study discussing model performance using core techniques? |

2.2.2. Data Extraction

In order to obtain the data which are needed to address our research questions and contributions, we used the data-extraction methods highlighted in Table 4.

**Table 4.** Data extraction.

| Study |
|---|
| Study Research Problem Contributions |
| RQ1: Heterogeneous Data |
| RQ2: Data Harmonization |
| RQ3: Industrial Textual Data and Techniques |
| RQ4: Sequential Data Techniques |

### 2.2.3. Information Synthesis

At this stage, the extracted data were aggregated to answer the research questions. For our research questions, we used the narrative synthesis method. Accordingly, we used tables and charts to present our results.

### 2.3. Report Review

Data extracted from the primary studies was used to answer our four research questions. The guidelines of References [29,30] were closely followed in the reporting of results.

### 3. Results

The summary of selected studies in the detailed arrangement is presented in Table A1 (Appendix A). A total of 70 studies were included in this review. Of those, 14 studies highlighted the RQ1, 25 studies covered data harmonization, 23 studies focused on the techniques, and 8 studies showed the performance measure of sequential text, as shown in Table 5.

**Table 5.** RQ studies.

| RQ | Studies |
| --- | --- |
| Heterogeneity | 14 |
| Data Harmonization | 25 |
| Industrial Textual Data | 23 |
| Sequential Data Performance | 08 |

Figure 3 illustrates the number of studies per year. There were a smaller number of studies in 2015, and then the related research studies grew in 2016/2017, such as 15 and 18, respectively. In 2018 and 2019, the number of studies found was 16 and 15, and in 2020, four studies found.

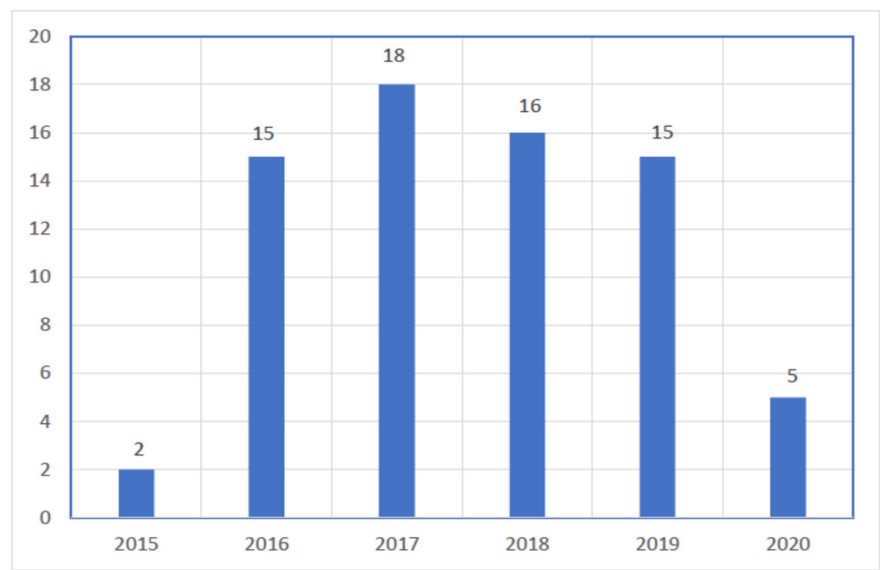

**Figure 3.** Studies selected per year.

Figure 4 illustrates the worldwide map of countries who contributed in research related to the research questions mentioned above. The research discussion on each research question is described in Table A1 (Appendix A).

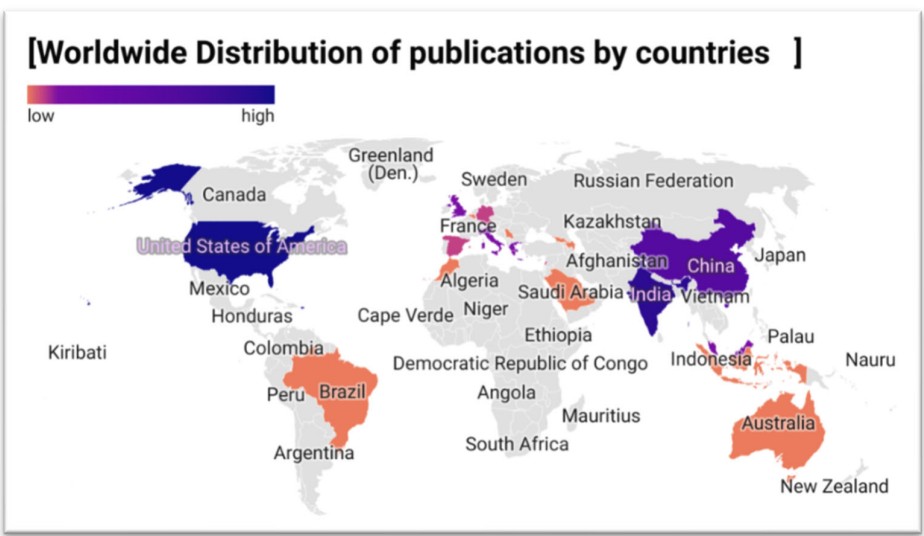

**Figure 4.** Chlorophet map showing the worldwide distribution of publications by countries.

### 3.1. Which of the Domains Are Mainly Focused on Researching Heterogeneity?

In this review, 14 studies discussed heterogeneity or heterogeneous datasets. Here we present a list of contributions for heterogeneous data.

Initially, the heterogeneity issues were presented by the researchers Silverio, Cavallo, De Rosa, and Galasso [31], in which the centralization of cardiovascular disease (CVD) is needed because no such system has been developed for all CVDs worldwide that will help patients and staff with responsible treatment. A framework was developed of all collected CVDs and compared based on performance ratio. In addition, a survey was conducted by a research team of Verma, Agrawal, Patel, and Patel [32] on the challenges faced by heterogeneous data in structured, semi-structured, and unstructured (SSU) formats. Various types of data are covered in SSU data, such as text, audio, images, video, and social media. In this, it is stated that Big Data Analytics (BDA) play an important role in Industrial Revolution 4.0, particularly for Big Data Analysts for making decisions, analysis, visualization, and prediction of challenges. The studies show the following challenges faced by industries: as predictive analysis, social media analytics, content-based analytics, text analytics, audio analytics, and video analytics. Moreover, Ali, Neagu, and Trundle [33] have highlighted in their work that heterogeneity has become an issue for large textual datasets used for data mining. For this, different machine learning (ML) techniques, such as KNN, SVM, NB, and ANN, are used for the classification of data, as well as for similarity, semantic-based information retrieval uses different algorithms. The result shows the accuracy of similar pairs to be better than existing classification techniques.

Likewise, the researchers Sivarajah, Kamal, Irani, and Weerakkody [34] emphasized that Big Data are growing rapidly in every type of their characteristics. Because of this issue, sometimes it is difficult to decide from large volumetric data. Multiple forms of interview forms, case studies, and experiment data files were used to check the assessment score and experts suggested that for timely decisions, data harmonization is needed, and it will remove heterogeneity from large data files. Likewise, in Bangalore city, a model of city bus heterogeneous data was proposed by Jaybal, Ramanathan, and Rajagopalan [35] to predict the bus timings, previous stops, and next stops on the map. The data formats were GPS, Web Data, CSV, and text formats. On the contrary, data heterogeneity of different Electronic Health Records (EHR), using deep learning (DL) techniques, was discussed by authors Shickel, Tighe, Bihorac, and Rashidi [36] to improve digitization of EHR. Chinese hospital data are used to create deep EHR projects with the help of DL

algorithms. Moreover, in Big Data (BD), massive data are produced in unknown and untuneful patterns, thus creating a data heterogeneity problem, as highlighted by Gheisari, Wang, and Bhuiyan [37]. In this study, different DL, ML, and BD techniques were highlighted to remove heterogeneity and make data useful.

Furthermore, data mining of heterogeneous data plays an important role in getting proper knowledge and information from huge datasets, as presented by Kalra and Lal [38] in their work. By using ML classification and clustering techniques, it is possible to fetch data. Moreover, the authors Kolhatkar, Patil, Kolhatkar, and Paranjape [39] expressed their views on online education systems, such as MOOC and Moodle. There are issues related to structured and unstructured data formats. It is difficult to manage and store heterogeneous data. To remove the heterogeneity issue, a conceptual model was proposed using software applications and RDBMS. In addition, the prominent authors Sindhu and Hegde [5] contributed with their work by proposing that the handling of large and complex data creates the problem of heterogeneity and is solved by using different BD techniques and text-mining algorithms. The conversion of unstructured into structured format performance is calculated in time. Moreover, real-time healthcare data are generated from sensors and gadgets in unstructured formats, which is not possible in structured formats with simple tools. Ismail, Shehab, and El-Henawy [40] stated in their work that healthcare data need to be monitored, visualized, and predicted time by time for these BD techniques, such as Hadoop and MapReduce, used for developing the EHR. Moreover, for the development of heterogeneity, Zhang et al. [41] highlighted the use of large-scale urban multisource heterogeneous data integrated by using tensor decomposition to solve the issues of urban town and to make it a smart town.

Additionally, Elsharkawy, Ahmed, and Salem [42] believe that the heterogeneity and complexity are two main issues highlighted for the solution and integration of clinical data. The data are generated in different formats, such as RDF, XML, and images. The issues were solved by using ontology and semantic techniques. Recently, the authors Arora and Goyal [3] emphasized that the various frameworks of heterogeneity and heterogeneous datasets can be used for solving the issues of heterogeneity, and it was shown that heterogeneity is always due to the unusual format of data and lack of integration resources, expertise, and techniques. Table 6 shows the advantages and disadvantages of studies selected for Research Question 1.

**Table 6.** RQ1 domains' advantages and disadvantages.

| Study Reference | Domain | Advantages | Disadvantages |
|---|---|---|---|
| [31] | Healthcare | Millions of E HR combined | Limited data access to selected smallest patient populations |
| [32] | General Purpose | diverse, massive, and complex data | Requires new norm of integration |
| [33] | Text similarity | Helps in classifying similar objects. Increase performance of ML techniques | It is considered inconsistent and ad-hoc. |
| [34] | General Purpose | The natural property of BD. Combine and manage | variety of inconsistent data create problems |
| [35] | Information Retrieval | Utilize data from a variety of sources. Single unified data source | Loss of information due to semantic, syntactic, and schematic difference |
| [36] | Healthcare | Electronic health record became unique | Difficult to manage different form |
| [37] | General Purpose | DL helps in solving the integration of heterogeneous data | Conflicting information shared by resources |
| [38] | Information Retrieval | Efficient data extraction Effectiveness of data | Meaningful data extraction from the huge database |
| [39] | Education | Helps in distribution across clusters | Need More memory for processing |

| [5] | Healthcare | Need to manage data efficiently and collaborative way | Need to be distributed and parallel computing systems and database |
| [40] | Healthcare | Multiple different sources data with a unified view | Data processing Speed and quality of data analytics |
| [41] | Infrastructure | Extremely large multisource infrastructure containing vehicles, residents, and smart card | Some models are only indirectly relevant to a particular phenomenon of interest |
| [42] | Healthcare | Semantic-based integration and semantic-based medical retrieval | Clinical records without linguistic standard |
| [3] | General Purpose | Heterogeneous data can be solved by using RDBMS,concept lattice, and MapReduce | Performance of all heterogeneous data is not calculated |

*3.2. How Does the Data Harmonization Resolve the Issues of Heterogeneity?*

In this section, 25 studies were selected which discuss data harmonization, data integration, and data fusion. The details of each study are discussed below.

Initially, heterogeneous oil and gas data are unorganized, which is difficult to manage. For that data harmonization was proposed by Danyaro and Liew [43], using semantic web and BD tools. Where performance of the precision, recall, and F-score found better than existing techniques. In addition, agriculture data are stored in clusters, and it is difficult to handle heterogeneous data. Therefore, a uniform format was reported by (Sambrekar, Rajpurohit, and Joshi [44], using Couchbase and NoSQL, and it was found that the time duration for fetching records is fast. Apart from this, different frameworks have been developed by different organizations to make decisions, but no framework has been proposed for value creation. In this study, Saggi and Jain [45] created a framework for value creation from SSU data also in-depth issues of heterogeneity, harmonization, and BD techniques were highlighted. It shows the importance of data integration for industrial data, decisions, product reviews, and visualization of future strategies. Artificial intelligence, ML, and cloud computing will be helpful for BD Analysts. Moreover, Li, Chai, and Chen [46] summarized that the heterogeneous data in industry are produced easily but are difficult to store, manage, and audit. In this study, the issue of heterogeneity of large firms was solved using a NoSQL-based data integration model. Furthermore, health data are very important for patient treatment, monitoring, and satisfaction. Health data are generated by all institutes by using open-source web data, but no such online module has been proposed for integration of all web-based centralized. In their study, Hong, Wang, et al. [47] revealed a Web-based FHIR visualization tool, using a standard structured format API. Again, Lopes, Bastião, and Oliveira [48] described that the file sharing between users was difficult for heterogeneous data. Therefore, a real-time integration and interoperability model was developed by using PostgreSQL to facilitate different users.

In addition, Yuan, Holtz, Smith, and Luo [49] mentioned that the child-patient disorder/condition data were complex and unmanageable due to manual work and human involvement. To overcome this issue, different preprocessing, NLP, and ML tools are used to create patient data in digital form and without any biases. The performance of the autism spectrum is calculated using precision and recall. Furthermore, Daniel [50] also emphasized on the issues and challenges faced by educational institutes and researchers are highlighted, such as data integration and sharing between campuses and branches. Besides this, text-free or unstructured data in healthcare data create issues for managing and storing. Therefore, data fusion was suggested by the Kraus et al. [10] to manage the heterogeneous data. Moreover, in an online learning system, data need to be integrated and efficient for smart educational systems. Data processing and storage of audio, video, images, and text formats was developed by Dahdouh, Dakkak, Oughdir, and Messaoudi [51] with the help of Hadoop, MapReduce, and Spark. As a result, it helps in taking a smart decision within seconds. Additionally, Patel and Sharma [52] explained the various issues

of data harmonization in this survey. Before that, data warehousing and OLAP were used, which do not support huge datasets of open source and unstructured formats. In the end, different BD and ML techniques are suggested for dealing with huge data. Consequently, in the oil and gas industry, data are generated in operational formats from different clusters at a time, which needs data integration to collect data in a centralized place for making timely decisions identified by Alguliyev, Aliguliyev, and Hajirahimova [53].

Wang [54] mentioned that the disparate data are generated in unstructured formats, such as sensors and text, which describe heterogeneous behavior. For this reason, a data integration model was developed to solve the technical and quality problems of BDA. The model was developed using ML and DL techniques so that BD analysts could visualize, analyze, and make decisions from disparate data. Additionally, Chondrogiannis et al. generated a tool for clinical data in a heterogeneous form and for integration of data, an ontology-based tool suggested to arrange data in a structured format. Moreover, patient cohort and biomedical data play an important role for previous health treatment and analysis, and data provided by patients in a heterogeneous structure need to be harmonized, as argued by Kourou et al. [11], so that, in an online tool, all patient data are available to medical staff during analysis. In this survey, different cohort harmonization techniques were highlighted, which will help in healthcare applications, such as ML, DL, and Ontology techniques. In addition, in an urban town, so many issues related to basic needs was mentioned by Souza et al. [55]; the objective of that study was to make the urban town into a smart urban town. Data are generated by different departments in JSON, string, and maps. To make smart decisions, all data must be integrated.

Furthermore, the patient stays in hospital data with different codes were not publicly available to make an health records into an EHR reported by authors (Scheurwegs, Luyckx, Luyten, Daelemans, and Van den Bulcke [56]. By using Naïve base and Random Forest on the UZA dataset, the patient classification was performed. Similarly, the researchers Jayaratne et al. [57], in their study, stated that the web-portal-based patient data produced by many healthcare hospitals in different formats were difficult to decide due to decentralization. To solve this issue, an automated and centralized web portal was developed which helps with online decisions. In contrast, the research team of Hong, Wen, Stone, et al. [58] analyzed that the patients with obesity and comorbidities were monitored after discharge from hospitals. The objective of this study is to develop a patient-centric system for FHIR using NLP toolkits and ML algorithms from the Mayo Clinic, MIMIC III, and i2b2 datasets. The overall performance of this system is measured in precision, recall, and F-Score. In addition, the same authors, Hong et al. [59] proposed a model for the quality and performance-based data integration for information extraction, using NLP, ML, and Bag of Words (BoW). Moreover, Hong et al. [60] used a Mayo Clinic dataset with the help of NLP toolkits for making a digital FHIR system. In contrast, Chen, Zhong, Yuan, and Hu [61] conducted a review and suggested a unified model for SSU data, using MapReduce. Besides that, XML-based OGOLOD datasets were accessed by using ontology tools for a semantic oriented data harmonization model that was presented by Carmen Legaz-García, Miñarro-Giménez, Menárguez-Tortosa, and Fernández-Breis [62].

In Saudi Arabia, patient health data generated in public and private hospitals are not shared and integrated with the health information system due to a lack of heterogeneity. Therefore, the Banu, Kuppuswamy, and Sasikala [63] team proposed a NLP and BDA-based systems. Lastly, online FHIR-based web portals were developed by using NLP techniques and open-source tools on the Mayo Clinic dataset to centralize the data generated in a heterogeneous format that was revealed by the researchers Hong, Wen, Shen, et al. [64]. The contributions of all studies in all domains are discussed in Table 7.

**Table 7.** RQ2 domain and contributions.

| Study Reference | Domain | Contributions |
|---|---|---|
| [43] | Oil and Gas | High performance measure |
| [44] | Agriculture | High performance, high availability, and high scalability, using the latest techniques |
| [45] | General-Purpose | Data generation, storing, fetching, analysis, visualization, and decision-making |
| [46] | Banking | Helps in auditing the multisource data |
| [47] | Healthcare | Facilitate for navigation of HL7 FHIR core resources |
| [48] | General-Purpose | Delivering automatic services to interoperable system |
| [49] | Healthcare | Helps in developing an automatic system for disordered patient |
| [50] | Education | To motivate researchers and academicians about the latest techniques |
| [10] | Healthcare | Useful for decisions of scientific, clinical, and administrative work |
| [51] | Education | Facilitate in online learning, storage, processing, and academic activities |
| [52] | General-Purpose | Recommendation system, opinion mining, and parallelism can be targeted |
| [53] | Oil and Gas | Helpful for decision-makers during exploration, drilling, and production |
| [54] | General-Purpose | It will facilitate for fetching data and performance measure |
| [65] | Healthcare | Helpful for disease prevention, tracking, and policy making |
| [11] | Healthcare | Helps in boosting statistical power of sustainable and robust data |
| [55] | Infrastructure | Geographic based smart city for aggregation, visualization, and analysis |
| [56] | Healthcare | Helps in predicting the clinical codes of patient stays |
| [57] | Healthcare | Helps in patient-centered care decision-making among stakeholders |
| [58] | Healthcare | Helps in finding the patient having obesity and comorbidities |
| [59] | Healthcare | Helps in developing patient diagnostic criteria and representation |
| [61] | General-Purpose | Support in integration, storage, computation, and visualization |
| [62] | Healthcare | Open biomedical repositories can be developed in semantic web formats |
| [60] | Healthcare | Normalizing and integration of structured and unstructured EHR data |
| [63] | Healthcare | Helps health information system to keep a record of patients' data |
| [64] | Healthcare | Helps in standardizing the clinical data normalization |

*3.3. Which Techniques Are Being Used for Solving the Harmonization Issue for Large Textual Datasets?*

In previous studies, SSU heterogeneous data were used in the form of text, images, audio, video, and social media formats. The BD and BDA literature reviews proposed so many models and frameworks for data harmonization or integration. Among them, textual data play an important role in semantic, syntactic, and schematic data from large datasets. In different industries, different approaches are used by BD analysts to meet the demands of users and owners.

In this section, 16 studies were selected that highlight the core techniques and their contributions in terms of performance, time, and accuracy in data harmonization, data integration, and data fusion. The details of each study are discussed below.

At first, Tekli [66] found that, in the entertainment industry, the feedback given by the audience in form of large sentences and getting semantic meaning from XML documents is very challenging. Additionally, Sanyal, Bhadra, and Das [67] pointed out that, by using business intelligence tool sentence-similarity retrieved, the technique proposed for the IT Ecosystem has been adopted by business firms. Apart from that, in the health sector, data are also important for harmonization, as noted by Adduru et al. [68]. They also discussed how the dataset contains many clinical codes and how it is difficult to get information and text classification to solve the issue. NLP techniques, such as N-Gram, Jaccard Similarity, Word2Vec, and different DL approaches, are used to create a paraphrasing dataset from clinical data. Similarly, the research team of Mujtaba et al. [69] revealed, in a clinical-text-classification review that the approaches for textual data play an important

role, especially in supervised ML techniques. Likewise, a medical prescription is a document of proof about a patient's health history recorded during the diagnosis, but sometimes it is difficult to understand the semantics of prescribed medicines was presented by Yanshan Wang et al. [70]. In this study, the Mayo Clinic dataset was utilized with the help of NLP techniques to find the semantic and similarity scores of medical texts. On the contrary, a study was proposed by Chen, Hao, Hwang, Wang, and Wang [71] that states that the healthcare communities manage healthcare data on web-based portals but are not available to all medical practitioners. For the prediction of chronic diseases, ML classification algorithms, such as CNN, NB, KNN, and DT, are used for analysis. Besides that, the authors Pathak and Lal [72] focused on open-source files-based heterogeneous datasets developed by using Modified IDF cosine similarity for information retrieval. A very detailed and descriptive survey was carried out by the authors Torfi, Shirvani, Keneshloo, Tavvaf, and Fox [73]. In this survey, different datasets of open-source NLP tasks, using different DL methods on BERT models, were discussed to summarize text and word embedding. In addition, Wu, Zhao, and Li [74] proposed that phrases of NLP models be vectorized by using the phrase2Vec model to overcome the issues of BoW and preprocessing. In the same way, the authors Moscatelli et al. [75] stated that patient data are very critical and sharing them is possible with high-security algorithms. By using NoSQL, MongoDB, and NLP techniques on XLS, CSV, and TXT files, data acquisition and simulation are possible. Similarly, Chen, Du, Kim, Wilbur, and Lu [76] also emphasized that, with the use of advanced technology, the health sector can be upgraded. Furthermore, health records can be in the digital form of clinical data and support multiple formats, but it is not easy to fetch similar data for digital records without the latest techniques in text mining. DL-based entities fetched from STS datasets combine rich features. Despite this, Mahlawi and Sasi [77] found that, from the large number of Enron email datasets, data are extracted by using NLP and sentiment analysis to make them available in a structured format. Furthermore, the authors Eke, Norman, Shuib, and Nweke [78] noted that the other parts of NLP are also important. In that, lexical analysis and ML-based emotional behavior detected from the text messages were used to check the level of criticism or hurt level from the Sarcasm dataset. Moreover, biomedical text mining was performed by using text preprocessing, clustering, classification, and information-extraction techniques mentioned by Allahyari et al. [79]. This led authors García, Ramírez-Gallego, Luengo, Benítez, and Herrera [80] to focus on Indian regional multilingual data processed with the help of natural language-processing techniques. Finally, Harish and Rangan [81] suggested that text be processed through ML and DL algorithms for semantics. BD processing for huge data is performed by using BD tools and libraries.

The contributions, techniques, and domains of all studies are discussed in Table 8.

**Table 8.** Domain, techniques, and contributions.

| Study Reference | Domain | Techniques | Contributions |
|---|---|---|---|
| [66] | Entertainment | Sentence Similarity | Using computational identification of the meaning of data in context from XML large datasets |
| [67] | Business | Business Intelligence | To extract values from the organized and refined data also helps stakeholders to make decisions |
| [68] | General | Sentence Similarity | Helps in training deep learning models for clinic paraphrase generation and simplification |
| [69] | General | Feature Extraction | To calculate the performance of expert-driven and fully automated features on free-text clinical reports |
| [70] | Healthcare | Sentence Similarity | Support reduction in cognitive burden and improvement in the clinical decision-making process |
| [71] | Healthcare | Word Embedding | Analysis of illness will help early disease detection, patient care, and community services |

| [72] | General | Information Retrieval | Document retrieval from large datasets of disparate formats |
| [73] | General | Deep Learning | Automatic semantic analysis and data drove strategies for Computer Vision, ASR, and NLP |
| [74] | General | Phrase Embedding | Helps in the integrity of semantic units and vectorization of similar words |
| [75] | Healthcare | Preprocessing | To precise historical analysis of clinical activities of patient |
| [76] | Healthcare | Sentence Similarity | To get clinic semantic textual similarities such as lexical patterns, word semantic, and named entities |
| [77] | Email | Preprocessing | Unstructured email extracted in a structured format |
| [78] | Social media | NLP | To find the sentiment from the message concerning content and level of hurting or criticizing |
| [79] | General, Health | Knowledge discovery Database | All tasks and techniques related to textual data, such as IR, NLP, IE, text summarization, unsupervised and supervised ML, opinion mining, and biomedical text mining, are discussed |
| [80] | General | NLP, BD tools | Data with long instances of text processed using tools and libraries |
| [81] | Multilingual | NLP, ML | Multilingual text preprocessing issues by using text mining and processing discussed |

*3.4. How NN Algorithms Are Well-Suited with Respect to Efficiency for Large Sequential Datasets*

In this section, 8 studies were selected which highlight the performance of Recurrent Neural Network (RNN) and Convolutional Neural Network (CNN) used for sequential data. The details of each study are discussed below.

At first, the researchers Yin et al. [82] and Ouyang et al. [83], in both surveys, discussed the use of NLP and DL techniques for fake-news detection and sequential data. By using techniques, it is found that the accuracy of model is up to 93%. Moreover, a comparison of CNN and RNN reveals that RNN is better than CNN. The techniques can be used for sentimental, relational, textual entitlement, answer selection, QA path query, and POS tagging was pointed by Lopez and Kalita [84]. Additionally, the authors Chai and Li [85] selected the studies that work for the Chinese community. In that, Chinese language based Clinical NER's performance was increased by using NLP techniques with DL. Similarly, the other techniques such as RNN with DL always shows better results which was presented by authors Oshikawa, Qian, and Wang [86]. In addition, with the help of NLP in the different domains, the sequential data performance is optimum also highlighted by Young, Hazarika, Poria, and Cambria [87,88]. lastly, a survey was conducted by the authors Jing and Xu Jing and Xu [89,90] which depicts the performance of RNN with the addition of NLP is at it shows the performance at its peak.

The contributions, techniques, and domains of all studies are discussed in Table 9.

**Table 9.** Model Performance Techniques.

| Study Reference | Domain | Techniques | Contributions |
| --- | --- | --- | --- |
| [82] | General | CNN, RNN for NLP | RNN perform better |
| [83] | Healthcare | RNN, N-Gram | RNN performance better by using N-gram |
| [84] | General | Compared with the existing Algorithm of CNN | RNN outperformed |

| [85] | General | Used in many NLP and audio-video functionality | Better for sequential text |
|---|---|---|---|
| [86] | Fake News | RNN for larger data sets of fake news | 93% accuracy |
| [87] | General | CNN, RNN | RNN is better as per recent studies |
| [88] | Cancer, healthcare | DL classifier is better than conventional classifier | Model accuracy is better by using RNN |
| [89] | General | FFNNLM, RNNLM | RNN Language model is best |
| [90] | Medical, General | CNN, DBN, RNN | RNN is better in terms of NLP |

*3.5. Common Techniques*

In this paper, different heterogeneous data and data harmonization approaches are discussed. The core techniques used in studies are presented in Table 10. The NLP technique is used all together in some studies, and also in some studies as a separate technique. For larger datasets, different techniques are highlighted that were used to help different domains. The major techniques include N-Gram, Bag of words, Bag of Phrases, TFIDF, cosine similarity, Jaccard similarity, Jaro Winkler, word2Vec, Phrase2Vec, Doc2Vec. It also helps with Name Entity Recognition (NER), text summarization, predictive analysis, word embedding, and semantic-based feature extraction from large heterogeneous datasets.

Along with NLP techniques, different machine learning algorithms for classification and clustering are highlighted in Table 10. Moreover, for training and testing of sequential data, deep learning algorithms are used for better performance and efficiency. Recurrent Neural networks performed better than CNN with the help of NLP and BD techniques. Other than that, BD, database, and web-based techniques are also discussed in different studies. For making structured formats, NoSQL, RDBMS, SQL, PostgreSQL, and ETL were used, and from the studies, it was found that the performance of structured data using these techniques is better. For web-based data, XML and ontology tools are used to fetch the data and place them in a structured format. Table 10 contains the core techniques of ML, DL, and NLP. Core techniques are also added that are based on studies relevant to research questions.

**Table 10.** Common techniques.

| Category | Techniques | Studies' References |
|---|---|---|
| Storage Technology | NoSQL | [3,44,46,53,67] |
| | HDFS | [4,5,40,51,53,67,75,80,91] |
| | PostgreSQL | [48] |
| | DWH | [53] |
| | OLAP | [53] |
| | ETL | [91] |
| | SQL | [91] |
| | RDBMS | [39] |
| Web-Based Processing Technology | Semantic web | [43,62,66] |
| | Ontology | [11,42,62,65] |
| | Web | [57,64] |
| | Open-source | [47] |
| Platform Technology | Couch base | [44] |
| | MapReduce | [4,51,53,61,67,80] |
| | Spark | [51,67,80] |
| Processing Technology | ML | [37,38,54,58,59,69,78,79,81] |
| | DL | [36,37,54,68,79,81] |
| | Tensor | [41,61] |

| | |
|---|---|
| KNN | [33,71] |
| SVM | [33,49] |
| NB | [33,56,71] |
| RF | [56] |
| DT | [71] |
| K-Means | [74] |
| ANN | [33] |
| CNN | [71,92] |
| RNN | [92] |
| | NLP | [5,49,58,60,63,70,74–77] |
| | N-Gram | [49,68] |
| | TFIDF | [49,72] |
| | Jaccard Similarity | [68] |
| | Word2Vec | [68] |
| | BoW | [59] |
| Information Processing Technology | Cosine Similarity | [72] |
| | Text Summarization | [73] |
| | Word embedding | [73] |
| | JaroWinkler | [93] |
| | Soft TFIDF | [93] |
| | Doc2Vec | [49] |
| | Text preprocessing | [79–81] |

## 3.6. Data Formats

In Table 11, data formats or sources of data are presented. Large heterogeneous datasets and representations are found in different formats. Based on these formats, structured, semi-structured, and unstructured data are categorized. The structured data formats that are used in the studies are SQL, XLS, and String. On the other hand, semi-structured data formats that are used are CSV, JSON, XML, URI, RDF, GIS, and GPS. Unstructured data formats are categorized as sensor data, video, audio, images, text, industrial data, and social media files.

**Table 11.** Data formats.

| Techniques | Studies' References |
|---|---|
| URI | [43] |
| RDF | [43,47] |
| Sensor | [40,44,54] |
| TXT | [5,35,44,75,79,93] |
| CSV | [35,44,48,75] |
| XML | [44,47,62,66] |
| SSU | [3,31,33,34,36,37,41,45,46,54,58,60,61,63,68–71,73,74,76,77] |
| GIS | [46] |
| JSON | [47,48,55] |
| SQL | [48] |
| OCR | [49] |
| INDUSTRIAL | [2,53,67] |
| VIDEO | [51,92] |
| AUDIO | [51,92] |
| IMAGE | [51,92] |
| WEB | [11,35,37,39,52,56,59,64,72,91] |
| TEXTUAL | [52] |

| | |
|---|---|
| XLS | [5,65,75] |
| GPS | [35] |
| STRING | [55] |

### 3.7. Performance Measure Techniques

The performance measure is important for all types of models, tools, techniques, and algorithms used in industrial projects and data-fetching mechanisms. Decision-making, visualization, and prediction are the key roles played by BD analysts and their role in making the system efficient and effective. In this paper, different types of performance are measured in different studies, which are shown in Table 12. Among all, precision, recall, and F-score are dominant in all types of BD-, ML-, DL-, and NLP-based models and frameworks. Accuracy is used by many researchers, where business intelligence tools are used only by business firms which was highlighted by [31].

Ratio comparison and similarity score are calculated in studies where there is a comparison of pairs or similar sentences. Correlation of terms used for linking different health records of similar codes can be found in Reference [76]. Time is used for conversion or fetching records from larger datasets [5,51]. Based on the performance measurement techniques, it is found that it will help BD analysts to use a data harmonization model, relevant tools, effective techniques, and efficient algorithms that are used for disparate nature of data domains and real-time applications.

**Table 12.** Performance measure techniques.

| Study Reference | Precision | Recall | F-1 | Accuracy | BI | Score | Correlation | Time | Ratio |
|---|---|---|---|---|---|---|---|---|---|
| [43] | ✓ | ✓ | ✓ | ✗ | ✗ | ✗ | ✗ | ✗ | ✗ |
| [66] | ✓ | ✓ | ✓ | ✗ | ✗ | ✗ | ✗ | ✗ | ✗ |
| [49] | ✓ | ✓ | ✗ | ✗ | ✗ | ✗ | ✗ | ✗ | ✗ |
| [51] | ✗ | ✗ | ✗ | ✗ | ✗ | ✗ | ✗ | ✗ | ✓ |
| [67] | ✗ | ✗ | ✗ | ✗ | ✓ | ✗ | ✗ | ✗ | ✗ |
| [31] | ✗ | ✗ | ✗ | ✗ | ✗ | ✓ | ✗ | ✗ | ✗ |
| [33] | ✗ | ✗ | ✗ | ✓ | ✗ | ✗ | ✗ | ✗ | ✗ |
| [68] | ✗ | ✗ | ✗ | ✓ | ✗ | ✗ | ✗ | ✗ | ✗ |
| [70] | ✗ | ✗ | ✗ | ✗ | ✗ | ✗ | ✓ | ✗ | ✗ |
| [34] | ✗ | ✗ | ✗ | ✗ | ✗ | ✗ | ✓ | ✗ | ✗ |
| [35] | ✗ | ✗ | ✗ | ✗ | ✗ | ✗ | ✓ | ✗ | ✗ |
| [55] | ✗ | ✗ | ✗ | ✗ | ✗ | ✗ | ✓ | ✗ | ✗ |
| [56] | ✗ | ✗ | ✓ | ✗ | ✗ | ✗ | ✗ | ✗ | ✗ |
| [71] | ✗ | ✓ | ✓ | ✓ | ✗ | ✗ | ✗ | ✗ | ✗ |
| [58] | ✓ | ✓ | ✓ | ✗ | ✗ | ✗ | ✗ | ✗ | ✗ |
| [59] | ✗ | ✓ | ✓ | ✓ | ✗ | ✗ | ✗ | ✗ | ✗ |
| [5] | ✗ | ✗ | ✗ | ✗ | ✗ | ✗ | ✗ | ✗ | ✓ |
| [41] | ✗ | ✗ | ✗ | ✓ | ✗ | ✗ | ✗ | ✗ | ✗ |
| [60] | ✗ | ✗ | ✓ | ✗ | ✗ | ✗ | ✗ | ✗ | ✗ |
| [42] | ✗ | ✓ | ✓ | ✓ | ✗ | ✗ | ✗ | ✗ | ✗ |
| [72] | ✗ | ✗ | ✗ | ✗ | ✗ | ✓ | ✗ | ✗ | ✗ |
| [73] | ✗ | ✗ | ✓ | ✓ | ✗ | ✗ | ✗ | ✗ | ✗ |
| [74] | ✗ | ✗ | ✓ | ✗ | ✗ | ✗ | ✗ | ✗ | ✗ |
| [75] | ✓ | ✓ | ✓ | ✓ | ✗ | ✗ | ✗ | ✗ | ✗ |
| [76] | ✗ | ✗ | ✗ | ✗ | ✗ | ✗ | ✗ | ✓ | ✗ |
| [64] | ✗ | ✗ | ✓ | ✗ | ✗ | ✗ | ✗ | ✗ | ✗ |
| [78] | ✓ | ✓ | ✓ | ✓ | ✗ | ✗ | ✗ | ✗ | ✗ |
| [93] | ✗ | ✗ | ✗ | ✗ | ✗ | ✗ | ✓ | ✗ | ✗ |

✓ Technique mentioned in the study. ✗ Technique is not mentioned in the study.

## 4. Discussion

The studies selected in this SLR are more from the medical and healthcare domains, discussing the heterogeneity issue and unstructured data. The main reason behind the heterogeneity issue is the heterogeneous data produced by healthcare devices, gadgets, and diagnostics reports. Due to the disparate nature of data, the need for integration or harmonization arises with the uniform representation of scattered data. In addition, textual data related to core techniques and their performance measurement were essential to be discussed in detail. The insight of this study mainly focuses on the type of SSU data; the industry focusing on the heterogeneity issue; the data harmonization or integration approach used by domains; and information-related core techniques, algorithms, tools, and models, and their performance assessment. Moreover, other aspects related to data harmonization and textual information retrieval were discussed to help the research community with up-to-date research related to textual data, such as conversion, integration, curation, and mapping of data, as well as text similarity, word embedding, sentence similarity, phrase embedding, and preprocessing techniques. All research questions, such as how harmonization solves the issues of heterogeneity, techniques, and performance, are discussed in Section B. Moreover, the recently implemented standard methods by researchers and data analysts for different purposes are shown in Table 10. The disparate nature of data produced in multiple formats, shown in Table 11, shows how commonly SSU textual data files are processed for desired output. The performance measurement technique discussed in Table 12 shows how techniques such as precision, recall, F-1, accuracy, and time will help researchers and managers to evaluate the performance of the core techniques, tools, algorithms, and models.

As we figured out by searching using different search databases, no such data harmonization SLR is proposed and presented that mainly focuses on textual data (SSU). Related work with data harmonization was performed by the research team at Stanford University in 2011. They proposed a multimodal deep learning technique for multimodalities to enhance the performance of audio/video networks in Reference [94]. With the advent of technology and tools, multimodalities were used by various researchers for big multimodal data by using text, audio, visual, and physiological signals [27]. The replacement of the data harmonization keyword was proposed, such as data integration and fusion [95]. A survey paper has been published that deals with affective computing [26] and related areas, such as opinion mining, emotion, and sentiment analysis. In that unorganized and unstructured data generated from consumer feedback is used to check the modalities' feasibility, physiological data helps them with anti-spoofing. Textual data are widely used for information retrieval, sentiment analysis, and text similarity, and a review was conducted on multimodal fusion for affective computing that mainly focuses on sentiment analysis in 2017. Text, audio, and video were data inputs, along with feature and decision format fusion techniques. Sentiment analysis was performed based on supervised and unsupervised approaches, and NLP, ML, and DL techniques were used. Meanwhile, textual modality, heterogeneity, and performance measurement of methods were unnoticed.

Later, different studies related to the proposed SLR were discussed. They are described below. In Reference [25], a multimodal machine learning survey characterizes the fusion modalities for multimodalities and issues such as representation, translation, alignment, fusion, and co-learning. Still, no parameter related to heterogeneity was discussed. In Kourou et al. (2018), different harmonization methods for healthcare case studies mainly concentrate on the percentage of data harmonized. Moreover, in Reference [96], sentiment was predicted with the help of the multimodal approach. Punctuation predicted from conversational speech, using semi-supervised multimodal fusion techniques, is presented in Reference [97]. The hierarchical fusion technique was used for sentiment analysis using TAF data and social images [98–100]. Moreover, a fake-news-detection framework was proposed in References [100,101] for Twitter and image multimodalities. In Reference [102], movie content similarity was detected by using text, auditory, and visual information. As in Reference [103], emotions with the recommendation of traveling

place framework proposed using DL techniques on text, pictures, and video data. The existing related work mainly focuses on analyzing single modalities other than textual data. Limited to core techniques, neither integration nor performance measurement is contrary to the proposed SLR.

Based on the literature review and results discussed above, many studies are presented, which focus on research related to research questions. The insight from these results will help in futuristic applications and domains such as zero-shot network, transfer learning, IIoT, Industry 4.0, and extended reality applications. These technologies produce data in bulk that are disparate, so the data need to be processed before storage and presentation. The techniques which fall under the umbrella of machine learning, deep learning, and NLP will be beneficial for the performance and extraction of information retrieval. Performance evaluation techniques such as precision, recall, F-1, accuracy, and time will help data analysts and managers in the selection of relevant tools and techniques.

### 4.1. Contributions

To the best of our knowledge, this is the state-of-the-art SLR that discusses heterogeneous textual datasets. The main objective of this study was to perform an in-depth review of the heterogeneity issue; the data harmonization or integration approach for massive and scattered information; computational techniques that can process and manage textual information in an efficient manner; and performance measurement of tools, techniques, and models. Previously, various related studies, such as multimodalities [25–27], discuss the disparate nature of data, but no study explains the issues stated in the research questions.

Based on the results and discussion, the knowledge delivered from this SLR will be helpful for researchers and data analysts who are dealing with the enormous types of industrial text files and datasets. As mentioned, the most recent machine learning, deep learning, and NLP techniques would be helpful in fetching, representation, and visualization of data that are in the form of multiple formats. In addition, performance measurement techniques will also assist in selecting optimal techniques, tools, models, and frameworks. Besides this, data harmonization will help with futuristic applications, such as IIoT, Industry Revolution 4.0, extended reality, and zero-shot network domain data.

### 4.2. Implications for Practice

Massive data produced by domains such as healthcare, banking, insurance, law, infrastructure, education, oil and gas, telecommunication, and entertainment were managed by managers and data analysts with existing tools and techniques. Textual datasets contain raw data and information; retrieving helpful information from applications such as user feedback, semantic similarity, textual similarity, word embedding, and emotion recognition is a challenging task for decision-makers. With the advent of the latest tools and technology, it is the responsibility of managers and data analysts to process, store, fetch, and efficiently represent data so that the decision-maker can perceive all the data and make decisions appropriately. To solve heterogeneity, heterogeneous data must be harmonized and presented so that effective decisions can be made. For uniform representations, managers and data analysts need to select the appropriate SSU data formats, such as Microsoft Excel Spreadsheet file (XLS), text file (TXT), JavaScript Object Notation (JSON), etc. Moreover, efficient, and effective machine learning, deep learning, and NLP techniques for textual data will help with faster training and testing approaches. As a result, the performance measure can be obtained through precision, recall, F-1, accuracy, and time.

Academically, data harmonization or data integration is a hot topic under the umbrella of Big Data variety, because it deals with the disparate nature of data and produces information in bulk. The information retrieval from SSU data formats needs time, and for

fetching the records, it needs to be uniformly represented. For this purpose, data harmonization can be an excellent approach to characterize, predict, and visualize. As for the consequences of academic performance, data harmonization is a domain where research can move forward.

## 5. Conclusions

Big Data describe an occurrence in the complex and dynamic growth of data, and it is challenging to manage the variety of data with simple tools and techniques. The main objective of this study was to review the heterogeneity issues, data harmonization approach, core techniques for textual data processing, and their performance measurement. Core techniques, such as NLP, ML, and DL, were applied to real-world applications and reviewed in detail, so that they can cover heterogeneity, harmonization, and sequential large textual datasets. It is assumed that the heterogeneity issue is solved by using data harmonization or data integration approaches for real-world applications. In addition, there is a requirement to adopt high-performance techniques, optimized algorithms, and efficient measurement techniques.

The main contribution of this study relates to Big Data variety and data analytics, in that it solves the most crucial issue of data heterogeneity: managers and data analysts. Before presentation or visualization, data must be uniformly managed and stored with the DH approach and the latest analytical techniques. Compared with existing related work, the multimodalities [25–27], specifically, focus on multiple data formats, such as text, image, audio, video, and visual representations. Along with this, information retrieval and classification techniques were used, but no such point was discussed as proposed in this study's research questions.

*Limitations and Future Research Directions*

The most important limitation of this SLR is data availability and context of the domain, as harmonization keyword uses in many applications. Another limitation is the biases in the selection of articles, SLRs, and surveys. Some relevant articles were not available, as the paper acceptance and publication phase was between 2015 and 2020. A minor limitation is the selection of studies in the English language.

Big Data and data analytics play an essential role in futuristic technologies, such as IIoT, extended reality, Industrial Revolution 4.0, transfer learning, and blockchain. Data harmonization can uniformly represent massive data produced by domains and application platforms for solving heterogeneity. Based on the detailed discussion and in-depth review of all the articles, techniques, tools, and models, the following are the essential suggestions for moving forward. In medical health, update the FHIR based EHR model by using NLP and DL techniques on SSU data. Information retrieval should be from domains such as emotion, affective computing, sentiment analysis, and content similarity from SSU datasets. Measurement of the effects of emotion recognition from textual data shows that real-time fusion methods can be used to fuse information extracted from raw data. It will also help in getting optimal solutions from clustering techniques and core NLP techniques. It will play an important role in the automation of the education system based on learners' choices and semantic meaning. Moreover, with Ontology and XML, DH can be used for web-based applications, models, and frameworks in healthcare, business, finance, education, oil and gas, and other industries, which will help them to retrieve information from users' semantic meaning and predictive analysis. Moreover, automatic text summarization, phrase similarity using bag of phrases, vocabulary generation, and the semantic meaning of long sentences using the DH approach will be performed for heterogeneous textual datasets. In addition, it will help users from any context, domain, and application.

**Author Contributions:** Conceptualization, G.K., S.B., A.A.I., and S.A.K.; methodology, G.K., S.B., A.A.I., L.F.C. and. S.A.K.; validation, G.K., S.B., A.A.I., and S.A.K.; formal analysis, G.K., S.B., and

A.O.B.; investigation; resources, G.K., S.B., A.A.I, A.O.B., and S.A.K.; data curation, G.K.; writing—original draft preparation, G.K. and S.B.; writing—review and editing G.K., S.B., A.A.I., A.O.B., L.F.C.; and S.A.K.; visualization G.K., S.B., A.A.I., A.O.B., and S.A.K.; supervision, S.B. and S.A.K.; project administration, S.B.; funding acquisition, S.B.,L.F.C. All authors have read and agreed to the published version of the manuscript.

**Funding:** This paper was partially supported with the funding of Universiti Teknologi PETRONAS, under the Yayasan Universiti Teknologi PETRONAS (YUTP) fundamental Research Grant Scheme (YUTP-FRG/015LC0297).

**Institutional Review Board Statement:** Not Applicable.

**Data Availability Statement:** Not Applicable.

**Conflicts of Interest:** The authors have no conflict of interest.

**Abbreviations**

| Abbreviation | Full Name |
| --- | --- |
| RFID | Radio Frequency Identification |
| AI | Artificial Intelligence |
| SLR | Systematic Literature Review |
| ML | Machine Learning |
| DL | Deep Learning |
| NLP | Natural Language Processing |
| RQ | Research Question |
| ACM | Association of Computing Machinery |
| IEEE | Institute of Electrical, Electronics Engineering |
| ISI | Institute for Scientific Information |
| KNN | K- Nearest Neighbors |
| SVM | Support Vector Machine |
| NB | Naïve Base |
| ANN | Artificial Neural Network |
| GPS | Global Positioning System |
| CSV | Comma Separated Value |
| MOOC | Massive Open Online Course |
| RDBMS | Realtime Database Management System |
| RDF | Resource Description Framework |
| XML | Extensible Markup language |
| EHR | Electronics Health Record |
| BD | Big Data |
| NoSQL | Not only SQL |
| FHIR | Fast Healthcare Interoperability Resource |
| OLAP | Online Analytical Processing |
| JSON | JavaScript Object Notation |
| XLS | Microsoft Excel Spreadsheet |
| TXT | Text |
| IR | Information Retrieval |

| IE | Information Entity |
|---|---|
| CNN | Convolutional Neural Network |
| RNN | Recurrent Neural Network |
| FFNNLM | Feed Forward Neural Network Language Model |
| RNNLM | Recurrent Neural Network Language Model |
| DBN | Deep Belief Network |
| TFIDF | Term Frequency Inverse Document Frequency |
| ETL | Extract Transform Load |
| URI | Uniform Resource Identifier |
| GIS | Geographic Information System |
| BoW | Bag of Words |

## Appendix A

This section comprises Table A1, which illustrates the selection of studies based on inclusion and exclusion process.

**Table A1.** Selected studies.

| Study | Research Problem | Outcome |
|---|---|---|
| (Y. Wang et al., 2020) [52] | The semantic textual similarity of clinical text | Clinical similar text selected based on semantic behavior by using STS and NLP |
| (Torfi, Shirvani, Keneshloo, Tavvaf, and Fox, 2020) [75] | DL methods used for tasks and models | Different NLP basic tasks, their application, and using DL domains highlighted to enhance the system with high performance |
| (Wu, Zhao, and Li, 2020) [76] | Advancement of BoW and NLP | Performance of NLP and BoW enhanced by BoP and P2V to solve the problem of phrase embedding |
| (Eke, Norman, Shuib, and Nweke, 2020) [82] | Classification of sarcasm using NLP | Level of hurt or criticism in message text classification and NLP techniques discussed |
| (Harish and Rangan, 2020) [93] | Processing of low resource languages using language processing task | Raw text processed using text preprocessing, ML, and NN |
| (Hong, Wang, et al., 2019) [35] | FHIR visualization design, development, and evaluation | Visualization tool developed for HL7 FHIR users to add profiling |
| (Daniel, 2019) [39] | Data integration and sharing educational data | Use of Big Data in education especially ontological, technical, ethics, privacy, and lack of expertise discussed |
| (Dhayne, Haque, Kilany, and Taher, 2019) [44] | Healthcare data integration techniques and techniques | Issues with healthcare data integration, heterogeneity, and heterogeneous data are discussed |
| (Silverio, Cavallo, De Rosa, and Galasso, 2019) [45] | Heterogeneity of high volume, wide variety, and speed | Public health improvement, drug surveillance, integrating pharmacy, data integrity, data security, and legal issues were discussed for CVD |
| (Chondrogiannis, Andronikou, Karanastasis, and Varvarigou, 2019) [49] | Clinical data uniformity | Ontology-based data harmonization of clinical data terms expressed in a common frame |

| | | |
|---|---|---|
| (Mujtaba et al., 2019) [51] | Key aspects of clinical text | Supervised ML-based clinical text classification, feature extraction, representation selection technique discussed with future suggestions |
| (Jayaratne et al., 2019) [57] | Data Integration of patient to take a decision | Sports injury data integration platform developed for patient-centered healthcare and clinical decision support |
| (Hong, Wen, Stone, et al., 2019) [64] | Phenotyping framework for FHIR | ML algorithms and discharge summaries of patient conversion developed using NLP on the i2b2 dataset |
| (Ismail, Shehab, and El-Henawy, 2019) [69] | Solution for healthcare heterogeneous data | Healthcare system proposed for collecting and managing data from different healthcare devices to store and manage using BD tools and techniques |
| (Maheshwari, 2019) [4] | Issues associated with a variety of SSU data | Large heterogeneous data storing, processing, and transferring discussed |
| (Arora and Goyal, 2019) [3] | The various framework developed for SSU data | Heterogeneity and dimensionality issues discussed for data visualization |
| (Hong, Wen, Shen, et al., 2019) [80] | To develop pipeline using HL7 fast health | FHIR based Clinical unstructured and structured data integration model developed for EHR data normalization using NLP toolkits |
| (Chai and Li, 2019) [87] | Information processing from video recognition | Advanced applications used for information process discussed using RNN and CNN |
| (Guan et al., 2019) [90] | Cancer patient progress from free text | Information extracted and classified using NLP, DL, and ML techniques from cancer patient data |
| (Jing and Xu, 2019) [91] | Word sequence calculation from large data | Using NLP and ML the sequence of words and vocab identified |
| (Sambrekar, Rajpurohit, and Joshi, 2018) [32] | Conversion of unstructured data is not properly performed | SSU data conversion in addition it will provide better efficiency, scalability, and performance |
| (Saggi and Jain, 2018) [33] | Value creation from the different perspective of BDA | An integrated framework for Big Data and value creation from data |
| (Kraus et al., 2018) [10] | Data harmonization which is not available in healthcare | Data integration framework proposed to cover data harmonization, semantic enrichment, and data analysis process for medical data |
| (Dahdouh, Dakkak, Oughdir, and Messaoudi, 2018) [40] | An integrated online learning system | The architecture of BD for online learning system proposed with storage, processing, benefits for professionals, students, and teachers discussed in detail. |
| (Ali, Neagu, and Trundle, 2018) [48] | Classification of heterogeneous data | By applying classifier and algorithm on pairwise similarity to enhance quality and performance |
| (Adduru et al., 2018) [50] | Clinical text simplification using deep learning | DL based clinical text simplification and paraphrasing dataset developed |
| (Kourou et al., 2018) [11] | Data harmonization of biomedical data and cohort | Data harmonization and integration cohorts discussed with open challenges of biomedical data |

| | | |
|---|---|---|
| (Jaybal, Ramanathan, and Rajagopalan, 2018) [54] | Semantic, syntactic, and schematic view of data | Bus fleet operations analysis, diagnosis, and improvement of schedules discussed, and operation cost reduction proposed |
| (Hong et al., 2018) [71] | FHIR based digital data system using NLP | Unstructured and structured healthcare data integrated by using NLP tool to form a mapping of similar codes of medication data |
| (L. Zhang, Xie, Xidao, and Zhang, 2018) [74] | Multisource fusion using DL techniques | Multisource heterogeneous data-based data fusion model proposed to solve the issue of heterogeneity |
| (Moscatelli et al., 2018) [77] | Patient data sharing is critical | Clinical data of the patient and their precise historical analysis framework developed by using ML and BD tools |
| (Q. Chen, Du, Kim, Wilbur, and Lu, 2018) [79] | To get similarity core between clinical notes | DL models discussed for clinical semantic textual data similarity. |
| (Prasetya, Wibawa, and Hirashima, 2018) [83] | Measurement of text similarity algorithm | Lexical and semantic similarity performance measured between pairs |
| (Oshikawa, Qian, and Wang, 2018) [88] | Problems with fake news generation | Performance of fake news datasets, the technique of NLP for identification of fake news discussed |
| (Young, Hazarika, Poria, and Cambria, 2018) [89] | DL models and methods for NLP | RNN role in NLP applications such as Information retrieval, summarization, and their performance highlighted |
| (S. Patel and Patel, 2018) [4] | Usage of ML algorithm and performance | Heterogeneous data types highlighted where RNN and CNN are used for information retrieval |
| (Danyaro and Liew, 2017) [31] | A large amount of data are unorganized | Semantic-based integration model for O&G |
| (J. A. Patel and Sharma) [41] | Data harmonization of various heterogeneous data | Data quality, scalability, heterogeneity, and efficiency highlighted for disparate nature of data generated in form of Big Data |
| (L. Wang, 2017) [45] | The technical and quality problem of BDA | Technical challenges of data value, data mining, ML, and DL methods discussed for disparate data |
| (Sivarajah, Kamal, Irani, and Weerakkody, 2017) [53] | BD challenges for technology | BD characteristics issues such as heterogeneity, process challenges, and different textual analytics techniques discussed |
| (Souza et al., 2017) [55] | Urban planning issues of smart city | Heterogeneous-data-type-based integration of data from multiple departments integrated to make smart city |
| (Shickel, Tighe, Bihorac, and Rashidi, 2017) [58] | Data heterogeneity of EHR using DL | DL Techniques discussed for EHR data for larger healthcare datasets |
| (Gheisari, Wang, and Bhuiyan, 2017) [59] | BD and DL challenges for research trends | Data analytics, semantic indexing, preprocessing, and data governance research problems highlighted for BD |
| (M. Chen, Hao, Hwang, Wang, and Wang, 2017) [60] | Effective prediction of chronic disease using ML algorithms | Incomplete medical data in Chinese chronic diseases detected using CNN from structured and unstructured data |
| (Klašnja-Milićević, Ivanović, and Budimac, 2017) [62] | Perspective trends for education learning using BD | BD tools based online educational framework proposed for research and professionals |

| (Kolhatkar, Patil, Kolhatkar, and Paranjape, 2017) [63] | How to store and manage Unstructured data | Student activities, sentiment analysis, and predictive analysis suggested for educational heterogeneous data scope |
|---|---|---|
| (Sindhu and Hegde, 2017) [5] | Handling heterogeneity among large data | Conversion of unstructured data into structured data using text mining and HDFS for clinical text |
| (Pathak and Lal, 2017) [73] | Information retrieval from heterogeneous data files | Vector space model developed for information retrieval from large documents using MIDF cosine similarity |
| (Banu, Kuppuswamy, and Sasikala, 2017) [78] | HIS for Saudi hospital data management | Data integration model proposed for hospitals and HIS for Saudi Arabia using NLP and BD |
| (Mahlawi and Sasi, 2017) [81] | Structured data extraction from email | Unstructured email conversion into a structured format for knowledge extraction using NLP and text mining approach. |
| (Yin, Kann, Yu, and Schütze, 2017) [84] | Performance of CNN and RNN against NLP | For sequential and text matching RNN perform better than CNN |
| (Ouyang, Li, Jin, Li, and Zhang, 2017) [85] | To find limitations and type of medical entities for CNER | Performance analysis of rich context information with the help of medical vocab and POS using DL and NLP |
| (Lopez and Kalita, 2017) [86] | Enhanced CNN performance | Latest trends, techniques, and application of NLP and DL highlighted with performance measure and type of datasets |
| (Allahyari et al., 2017) [92] | Useful information extraction from the large volume of data | Text mining approaches, text preprocessing, clustering, classification, information extraction techniques discussed in detail with justification |
| (Tekli, 2016) [37] | XML based semi-structured semantic analysis | Textual data presented with a focus on semi-structured XML and ongoing challenges for XML disambiguation, semantic meaning, and combination |
| (Yuan, Holtz, Smith, and Luo, 2016) [38] | To make a digital system by converting unstructured and semi-structured data useful | By using NLP and ML information extracted from medical forms of ASD patients and result evaluated by experts with 91% recall |
| (Bhadani and Jothimani, 2016) [2] | Advancement in Big Data and web.2.0 | The latest tools, sources of Big Data, techniques, software applications, and technical limitations were discussed in detail |
| (Sanyal, Bhadra, and Das, 2016) [42] | Data processing using BD technologies | A conceptual model proposed for value creation and decision-making from huge Big Data |
| (Alguliyev, Aliguliyev, and Hajirahimova, 2016) [43] | Integration of transactional data and business analytics | Industrial data integration model proposed for large transactional data and visualization |
| (Verma, Agrawal, Patel, and Patel, 2016) [47] | SSU related issues | Predictive, social, text, audio, video analytics related issues were suggested for different industries |
| (Scheurwegs, Luyckx, Luyten, Daelemans, and Van den Bulcke, 2016) [56] | Structured and unstructured patient data are not accessible | The patient stays clinical code in structured and TXT format integrated to predict the type of isolation |

| (Kalra and Lal, 2016) [61] | Research challenges of heterogeneous data | Data-mining techniques discussed for SSU data produced in different formats to overcome the issue of data heterogeneity |
|---|---|---|
| (Hong et al., 2016) [65] | To convert textual data into the structured format | With the help of NLP and ML techniques, QDM developed for clinical diagnostics report will help in standard criteria development |
| (Z. Chen, Zhong, Yuan, and Hu, 2016) [66] | To develop a universal model to represent visual analysis | IBD model proposed which facilitates representation, management, and visualization. |
| (del Carmen Legaz-García, Miñarro-Giménez, Menárguez-Tortosa, and Fernández-Breis, 2016) [67] | By using semantic web-based data integration | Semantic web-based online and open dataset for biomedical data created using the integration of XML data |
| (Anagnostopoulos, Zeadally, and Exposito, 2016) [68] | Classification of 4v's of BD | BD integration framework discussed to highlight the challenges, tools, and techniques which will help stakeholders |
| (D. Zhang et al., 2016) [70] | Heterogeneity of urban cyber-physical system | Disparate nature of data in China integrated using the cyber-physical system to facilitate the urban system with 29% better performance |
| (Elsharkawy, Ahmed, and Salem, 2016) [72] | Semantic-based integration and information retrieval | Semantic-based health data integration and to enhance the performance of precision medicine |
| (García, Ramírez-Gallego, Luengo, Benítez, and Herrera, 2016) [94] | Processing of huge data analysis | Preprocessing techniques and libraries highlighted for huge data |
| (Li, Chai, and Chen, 2015) [34] | Heterogeneous data sources integration | A synchronized business audit data integration model developed using middleware technology |
| (Lopes, Bastião, and Oliveira, 2015) [36] | Automate real-time data integration | A platform provided for data and service for original data sources |

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
