# Peer review of "Data Harmonization for Heterogeneous Datasets: A Systematic Literature Review"

_applsci, doi:10.3390/app11178275_

Round 1

Reviewer 1 Report

The paper is well organized and understandable.

Some detailed suggestions are as follows:

1) In Figure 3, there should have a better data display and appropriate legend.

2) Combined with table 13, the author should elaborate on the specific expression of each metric and explain the significance of the metric in the DH task.

3) In Common Techniques, the author should distinguish processing technology (CNN/RNN/Kmeans), storage technology (HDFS/PostgreSQL), platform technology (spark/MapReduce), etc. again, or distinguish according to tasks (word2vec/word embedding/doc2vec can be classified into one category).

Author Response

Respected Reviewer, 

Please see the attachment attached. 

Reviewer 2 Report

Reviewer report

Dear Author(s):

In accordance with the review of the article "Data Harmonization for Heterogeneous Datasets: A Systematic Literature Review - (applsci-1364250)" which has been assigned to my person, I communicate below what my decision was:

Synopsis of the review

In terms of overall, the author(s) have done a good and innovative job. However, there are certain flaws that do not recommend the publication of the article in its current state. These flaws are especially centered on the presentation, data sources, data implementation, and contextualization of the manuscript. 

I list them:

  1. First of all, I have to ask you the typical question of why you have explicitly chosen these databases and not others (IEEE - Explorer, Science Direct, Springer, and ACM Digital Library). Please give in the paper an elaborated justification for this.
  2. Figure 2 is somewhat ambiguous. Please redo it and include a legend about each of the publications analyzed. Otherwise, a prospective reader would not get anything clear. Also, in the figure, please state "Years and number of publications".
  3. The breadth of table 6 requires you to make a number of changes. First of all, arrange the publications in descending order (not as it is now) to follow a certain coherence and to make the reading more friendly to a prospective reader. Align each column to the left and, above all, a problem I notice throughout the work, include the name of each of the authors before the number in square brackets. (Please do this for all references). Once this is done, move the table to a new section entitled appendices.
  4. Regarding the graphs, I think you need to include a new one that emphasizes the transnational aspect of the study. Please include a chlorophet map showing the worldwide distribution of publications by country.
  5. Between lines 242 and 460, the authors do not include purely an elaborated text but practically a numerical list of works. Unfortunately, I do not consider this to be appropriate. Please cite correctly by adding the name of the authors where appropriate and, of course, use grammatical connectors to link each of the publications reported.

  1. The discussions and conclusions are quite valid. Regarding the limitations that the authors point out concerning English language papers, can you give an indicative number of the papers that have been discarded because they were not written in English?

Once again, I would like to insist to the authors that they have done such great work.

Best regards,

The reviewer

Author Response

Respected Reviewer,

Thank you. 

Round 2

Reviewer 2 Report

On my part, I can only congratulate the auditors for the work they have done. Given that they have carried out all the changes and proposals for improvement proposed by this reviewer, it should not take long to answer. In my opinion, the manuscript should be accepted for its final publication.

With my best wishes for your academic and personal life,

The reviewer